# Schwannoma of the Hypoglossal Nerve Mimicking Carotid Body Paraganglioma

**DOI:** 10.3390/diagnostics12092122

**Published:** 2022-08-31

**Authors:** Maximilian Linxweiler, Wolfgang Reith, Mathias Wagner, Jan Philipp Kühn, Bernhard Schick

**Affiliations:** 1Department of Otorhinolaryngology, Head and Neck Surgery, Saarland University Medical Center, 66421 Homburg, Saar, Germany; 2Department of Diagnostic and Interventional Neuroradiology, Saarland University Medical Center, 66421 Homburg, Saar, Germany; 3Department of General and Surgical Pathology, Saarland University Medical Center, 66421 Homburg, Saar, Germany

**Keywords:** hypoglossal nerve, schwannoma, carotid body paraganglioma

## Abstract

Carotid body paragangliomas (CBPs) clinically present as highly vascularized cervical masses with a pathognomonic localization at the carotid artery bifurcation. Following ultrasonography and MRI/CT imaging, surgical resection with optional preoperative embolization is considered as the treatment of choice in most cases. We herein present the case of a 60-year-old female with characteristic clinical signs and imaging findings of a right-sided CBP who finally went to surgical treatment. Intraoperatively, the tumor showed an adherent growth to the hypoglossal nerve that had to be partially resected, resulting in a postoperative nerve palsy. Histological examination of the resected tumor revealed the unexpected diagnosis of a hypoglossal nerve schwannoma. To the best of our knowledge, we herein present the third case reported in the literature of a unilateral hypoglossal schwannoma located at the carotid bifurcation mimicking clinical symptoms, imaging and intraoperative findings of a CBP.

## 1. Introduction

Carotid body paragangliomas (CBPs), also known as chemodectomas or carotid glomus tumors, represent rare neuroendocrine tumors and account for about 0.5% of all head and neck neoplasms [1]. CBPs originate from paraganglionic tissue located in the adventitia of the internal carotid artery next to the carotid bifurcation. Physiologically, these cells function as sensors for blood arterial oxygen partial pressure, arterial carbon dioxide partial pressure, and arterial pH changes [2]. Sporadically, due to chronic hypoxemia or driven by genetic predisposition, CBP can arise from these cells, presenting as a slow-growing cervical mass. Over recent years, next-generation sequencing and gene expression profiling uncovered essential genetic features underlying the molecular carcinogenesis of paragangliomas [3]. It was shown that *SDH*, *VHL*, and *TMEM127* are the most frequently altered genes and can thus be characterized as tumor drivers in this entity. Depending on the tumor size, compression of adjacent structures, and potential catecholamine production, clinical symptoms can comprise a pulsatile cervical mass, palsy of cranial nerves IX/X/XII and difficult-to-control hypertension [4]. Imaging modalities for clinical diagnosis include ultrasonography of the neck, CT, MRI, and digital subtraction angiography (DSA) where CBPs present as highly vascularized, round-shaped tumors with a pathognomonic localization at the carotid bifurcation [5,6]. Depending on the tumor size, relationship with neighboring vessels and surgical challenge, CBP can be classified as Shamblin Type I, II, or III tumors [7].

Treatment options include surgical resection with optional preoperative embolization, conservative imaging follow-up, and radiotherapy [4,8]. The choice of treatment depends on several factors, e.g., tumor size, risk of surgical complications, patient age and comorbidities, multicentricity, and the need for vascular reconstruction. Major complications of CBP surgery include cranial nerve injury in about 25% and stroke in about 3.5% of cases [9,10].

## 2. Case Presentation

A 60-year-old Caucasian female presented at a tertiary care medical center with a three-month history of intermittent visual disturbance, bilateral weakness of arm elevation, and chronic headache. Past medical history included hypertension treated with Ramipril, cervical spondyloarthrosis, and latex allergy. After initially consulting her family doctor, an MRI scan of the head and neck was performed, showing a 2.6 × 2.1 cm measuring round-shaped tumor at the level of the right carotid bifurcation with an irregular but strong contrast enhancement as an incidental finding (see Figure 1a). The tumor showed a direct contact to the internal as well as external carotid artery, but with no signs of vascular compression or encasement.

Clinical examination of the patient showed no signs of cranial nerve palsy, laboratory tests including catecholamine blood levels, urinary metanephrines, and vanillylmandelic acid showed no pathological findings. Ultrasonography of the neck revealed a 2.6 × 2.1 × 1.3 cm tumor at the bifurcation of the common carotid artery with splaying but not compressing the branching internal and external carotid artery (see Figure 2). The tumor showed a strong vascularization in duplex sonography and a predominantly hypoechoic signal.

The clinical diagnosis of a right-sided carotid paraganglioma was made, and treatment options were discussed with the patient including surgery, radiotherapy, and periodic imaging follow-up. Based on the comparably small size of the tumor that was classified as Shamblin Type I being associated with a relatively low risk for surgical complications tumor resection with preoperative embolization was planned. Twenty-four hours before surgery, the patient was transferred to the radiology department for digital subtraction angiography and embolization. Herein, only small-sized feeding vessels originating from the internal carotid artery were found that were not accessible for selective catheterization and subsequent embolization (see Figure 1b). Nonetheless, the patient was taken into surgery the next day. Intraoperatively, the tumor was carefully dissected from the adjacent carotid artery branches that were not encased or infiltrated. In its cranial portion, the tumor showed a direct contact to the hypoglossal nerve with spreading of the nerve fibers. For complete tumor removal, several hypoglossal nerve fibers had to be sacrificed by sharp dissection. Postoperatively, the patient showed a palsy of the right hypoglossal nerve with deviation of the tongue to the ipsilateral side, which slightly impaired her speech and swallowing. With no further postoperative morbidities and uncomplicated wound healing, the patient left the hospital three days after surgery.

Histopathological examination of the resected tissue showed an in toto resected tumor surrounded by a fibrous capsule and being composed of vascular structures, regressive fibrous tissue, and predominantly spindle-shaped cells. Immunohistochemical staining showed positive reactions of these lesional cells for neuron-specific enolase (NSE), CD56, S100, vimentin, Sox10, and collagen type IV with negativity for P14, P63, chromogranin A, synaptophysin, glial fibrillary acidic protein (GFAP), desmin, melan A, HMB45, CK20, and AE1/3 (see Figure 3).

Based on histological architecture and immunohistochemical analysis, the diagnosis of a schwannoma was made, which was confirmed by an external pathology laboratory consulted for a second opinion. In a follow-up examination four weeks after surgery, the patient presented with an inconspicuous cervical scar and improved hypoglossal nerve function under ongoing physical and speech therapy. To exclude a schwannomatosis as a pre-disposition to develop further schwannomas, the patient was tested for *NF2*, *SMARCB*, and *LZTR1* gene mutations with no suspicious findings.

## 3. Discussion

To the best of our knowledge, we present herein only the third case reported in the literature of a hypoglossal schwannoma located at the carotid bifurcation mimicking clinical symptoms, imaging and intraoperative findings of a CBP.

Schwannomas are benign peripheral nerve sheath tumors and predominately affect the vestibular nerve in about 90% of cases [11]. Of the remaining non-vestibular schwannomas, hypoglossal nerve schwannomas represent only 5% of cases with less than 150 cases having been reported so far [12]. Of these cases, the vast majority arise from either the intracranial or the peripheral intralingual segment of the nerve [13]. Schwannomas of the cervical segment of the hypoglossal nerve presenting as mass in the parapharyngeal space are exceedingly rare with less than 40 cases having been reported in the literature to date [12,14]. When originating from the segment where the hypoglossal nerve crosses the carotid neurovascular bundle, tumor growth can lead to a spreading of external and internal carotid artery. Additionally, schwannomas can present with radiological characteristics comparable to CBP [13,15,16] so that, under these circumstances, peripheral hypoglossal schwannomas can anatomically, clinically, and radiologically closely mimic carotid body paragangliomas. To date, only two cases of hypoglossal schwannomas located at the carotid bifurcation have been reported [12,17], and we herein present the third case in the English literature.

Surgical treatment of carotid body tumor bears a relevant risk of cranial nerve injury that occurs in about 25% of patients, with hypoglossal nerve being the most common nerve injured [10]. The same applies to symptomatic schwannomas, i.e., tumors with local compression signs and/or affected nerve function where surgery remains the treatment of choice due to relative radioresistance. However, in cases of asymptomatic schwannomas with non-affected nerve function, expectant management with regular radiological and/or sonographic controls is usually recommended [18,19]. In our case, an at least partial resection of the hypoglossal nerve was unavoidable due to the direct anchorage of the nerve to the tumor with a spreading of nerve fibers. However, the patient presented in this report partially recovered hypoglossal nerve function under speech and physical therapy, which underlines the importance of preserving nerve fibers not directly attached to the tumor tissue. Seen retrospectively, one has to discuss whether partial sacrifice of the hypoglossal could have been avoided by expectant rather than surgical management when carefully evaluating all pre-operative diagnostic findings and potentially extending diagnostic work-up. In the presented case, ultrasonography and MRI showed typical findings of a carotid body paraganglioma and only DSA could have indicated a different histological entity. Surgical biopsy was not an option due to potentially severe bleedings in case of a paraganglioma. As we did not expect relevant additional information from further imaging approaches, we decided in favor of a tumor resection, which was also desired by the patient.

Taken together, this singular and illustrative case shows that hypoglossal schwannoma should remain a consideration in the evaluation of tumors located at the carotid bifurcation. As this report demonstrates, clinical and radiological as well as intraoperative findings can closely mimic that of carotid body paragangliomas and should therefore be considered as a differential diagnosis.

## Figures and Tables

**Figure 1 diagnostics-12-02122-f001:**
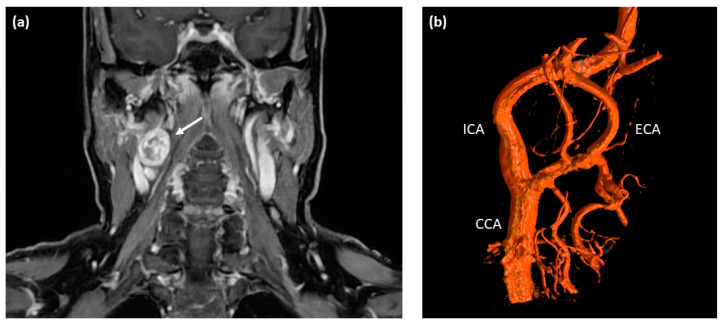
Pre-operative imaging of the right-sided cervical mass. (**a**) T1-weighted contrast-enhanced MRI scan of the neck depicting a round-shaped tumor located at the level of carotid bifurcation with irregular contrast enhancement (white arrow). (**b**) Digital subtraction angiography (DSA) showing a splaying of the internal and external carotid artery around the cervical mass located at the level of the carotid bifurcation; no major feeding vessels were detected making this tumor unsuitable for embolization. ICA—internal carotid artery; ECA—external carotid artery; CCA—common carotid artery.

**Figure 2 diagnostics-12-02122-f002:**
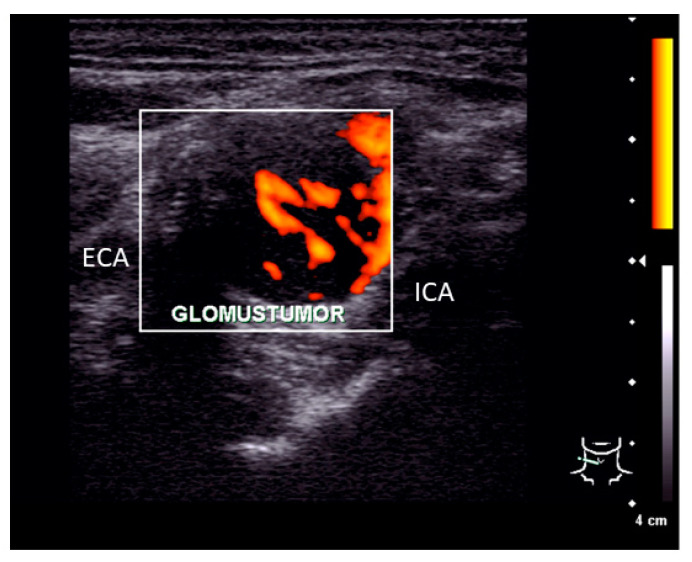
Ultrasonography finding. Pre-operative ultrasonography examination showed a 2.6 × 2.1 × 1.3 cm measuring tumor located between the internal (ICA) and external carotid artery (ECA) with a strong vascularization.

**Figure 3 diagnostics-12-02122-f003:**
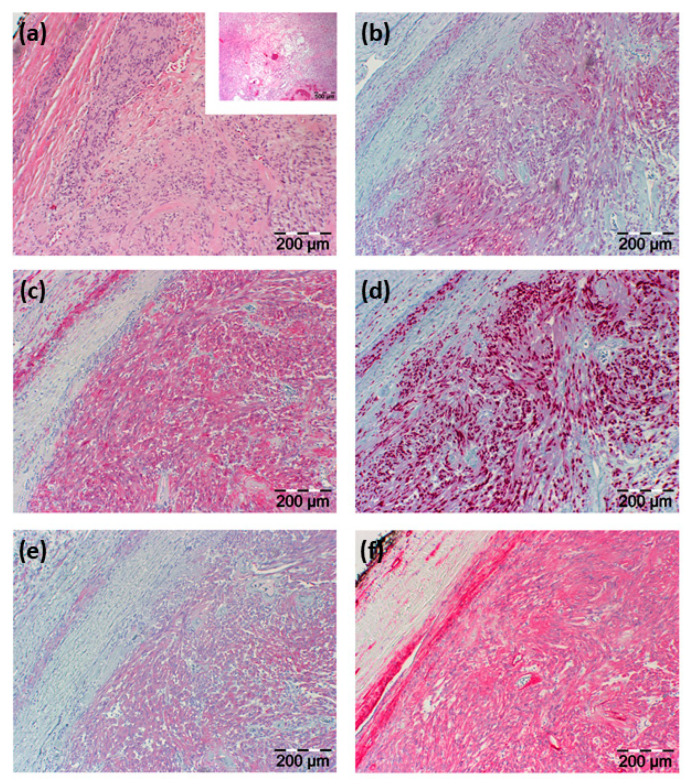
H&E histology and immunohistochemical staining results. (**a**) H&E staining illustrated in two different magnifications; the smaller image with a lower magnification shows regressive areas of the tumor; in (**b**–**f**) immunohistochemical staining is shown directed against (**b**) NSE, (**c**) S100, (**d**) Sox10, (**e**) CD56, and (**f**) collagen type IV. Section sizes are indicated by a scale bar.

## Data Availability

Not applicable.

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
