# Peer review of "Schwannoma of the Hypoglossal Nerve Mimicking Carotid Body Paraganglioma"

_diagnostics, 2022, doi:10.3390/diagnostics12092122_

Round 1
Reviewer 1 Report
Dear Editor,
I have read the paper "Schwannoma of the hypoglossal nerve mimicking carotid body paraganglioma". The authors describe an unexpected differential diagnosis of a carotid body paraganglioma that resulted in a schwannoma.
The paper is well written and has good iconographic material. However, I have two significant concerns.
The diagnosis of schwannoma is complex and challenging. CT scanning may not reliably distinguish a carotid body tumor from a schwannoma , although lack of flow voids with contrast enhancement may suggest schwannoma. MRI/MRA or angiography is commonly indicated to establish the diagnosis; I wonder whether the absence of related carotid body paraganglioma with the specific results of MRI and angiography presented should have suggested the differential diagnosis in advance. Expectant management of Schwannomas is typically the best strategy in the absence of neural deficit, so a more accurate differential diagnosis could have probably avoided the post-surgery side effect. Secondly, although isolated non-syndromic schwannomas are more common than multiple schwannomas (schwannomatosis), I would suggest that schwannomatosis should at least be ruled out. In order for it to be suitable for acceptance, the authors should at least comment on these two points.
Last comment: the keywords list contains neurofibroma which is not appropriate, schwannoma is not mentioned in the list; these two are not synonymous and they are two separate entities.
Reviewer 2 Report
The authors described the case of rate neurinoma of the hypoglossal nerve miicking a glomus tumour. The authors describe all of signs and publications focused on glomus tumor. Onlz a rare reference describe the neuronoma on the neck.
this diagnose neck neutinoma is more rare to gloms tumour generally. The problems is that the inteoduction associate that the generall problem is a glomus tumour. The introdution is focused on the flomus tumour. The readers will confused becouse the main diagnose is neurinoma. Tha same problems is oresentation of the diagnostic ser. The absence of relevant diagnostic protocol wich ifnore diturbancz between MR and USH beed improving. The any other methods for diagnose were not used and it need an explanation. The disscusion ignore the definitive histology and the analyse of mistakes in diagnose apprach and treatment as well. The rewritinig is the good idea.
Round 2
Reviewer 1 Report
Thank you for accepting the suggestions , I’ve not further comments.
Reviewer 2 Report
The te t was impoved a liitle. No all facts are described abd exlained well. The new facts about the genes were reduced a lot please see
A systematic review on the genetic analysis of paragangliomas: Primarily focused on head and neck paragangliomas http://dx.doi.org/10.4149/neo_2018_181208N933
